# vtRNA2-1/nc886 Produces a Small RNA That Contributes to Its Tumor Suppression Action through the microRNA Pathway in Prostate Cancer

**DOI:** 10.3390/ncrna6010007

**Published:** 2020-02-20

**Authors:** Rafael Sebastián Fort, Beatriz Garat, José Roberto Sotelo-Silveira, María Ana Duhagon

**Affiliations:** 1Laboratorio de Interacciones Moleculares, Facultad de Ciencias, Universidad de la República, Montevideo 11400, Uruguay; 2Departamento de Genética, Facultad de Medicina, Universidad de la República, Montevideo 11800, Uruguay; 3Departamento de Genómica, Instituto de Investigaciones Biológicas Clemente Estable, Montevideo 11600, Uruguay; 4Departamento de Biología Celular y Molecular, Facultad de Ciencias, Universidad de la República, Montevideo 11400, Uruguay

**Keywords:** vault RNA, vtRNA2-1, nc886, hsa-miR-886-3p, cancer, prostate, small RNA, microRNA, TCGA, PRAD

## Abstract

vtRNA2-1 is a vault RNA initially classified as microRNA precursor hsa-mir-886 and recently proposed as “nc886”, a new type of non-coding RNA involved in cancer progression acting as an oncogene and tumor suppressor gene in different tissues. We have shown that vtRNA2-1/nc886 is epigenetically repressed in neoplastic cells, increasing cell proliferation and invasion in prostate tissue. Here we investigate the ability of vtRNA2-1/nc886 to produce small-RNAs and their biological effect in prostate cells. The interrogation of public small-RNA transcriptomes of prostate and other tissues uncovered two small RNAs, snc886-3p and snc886-5p, derived from vtRNA2-1/nc886 (previously hsa-miR-886-3p and hsa-miR-886-5p). Re-analysis of PAR-CLIP and knockout of microRNA biogenesis enzymes data showed that these small RNAs are products of DICER, independent of DROSHA, and associate with Argonaute proteins, satisfying microRNA attributes. In addition, the overexpression of snc886-3p provokes the downregulation of mRNAs bearing sequences complementary to its “seed” in their 3′-UTRs. Microarray and in vitro functional assays in DU145, LNCaP and PC3 cell lines revealed that snc886-3p reduced cell cycle progression and increases apoptosis, like its precursor vtRNA2-1/nc886. Finally, we found a list of direct candidate targets genes of snc886-3p upregulated and associated with disease condition and progression in PRAD-TCGA data. Overall, our findings suggest that vtRNA2-1/nc886 and its processed product snc886-3p are synthesized in prostate cells, exerting a tumor suppressor action.

## 1. Introduction

Prostate cancer (PrCa) is a heterogeneous disease, the molecular mechanisms of which are still not fully elucidated [1]. It is currently the solid tumor with the highest incidence in men in Western countries, representing the second leading cause of male cancer death [2]. The incidence of PrCa is increasing mainly because of population ageing, increased awareness, and the widespread introduction of the prostate-specific antigen (PSA) test. Despite the fact that most patients can be successfully treated, a minor proportion develop an aggressive form of the disease that is currently incurable. Despite the remarkable progress in the development of molecular biomarker to aid disease diagnosis and risk stratification [3], improved biomarkers for treatment prediction and patient surveillance are still needed. The current advance of genomic technologies has contributed to the identification of thousands of non-coding RNAs (ncRNAs) in the human genome; nonetheless, no function has been assigned for the majority of them [4]. NcRNAs have emerged as significant players in cancer initiation and progression [5], therefore their clinical value is under intense investigation [6]. Pre-miR-886 was annotated as a microRNA precursor in 2007 (hsa-mir-886) (release v10 of miRBase (08/2007)) [7] after the identification of microRNAs derived from it in small RNA libraries from different human cell types [8]. Due to the recognition of its sequence homology with the human vaults RNAs (vtRNAs) vtRNA1-1/2/3 [9], pre-miR-886 was later re-classified as a vault RNA (vtRNA2-1) and therefore removed from miRBase (release v16 (11/2011)). Although the vtRNAs were discovered for their association with the vault particle [10], there is a large cytoplasmic fraction of vtRNAs, including vtRNA2-1, not associated with it [11,12]. Recently, vtRNA2-1 was proposed as a new type of ncRNA functioning as a tumor suppressor that inhibits Protein Kinase RNA-activated (PKR), and was consequently renamed as “nc886” [12,13,14,15]. In agreement with the former result, we previously found that vtRNA2-1/nc886 (from here on referred as nc886) is a tumor suppressor in the prostate, the expression of which is epigenetically silenced in tumors and associated with a worse clinical behavior [16]. In addition, its overexpression leads to a decrease in in vitro cell proliferation and invasion and a decrease in in vivo tumor growth [16]. Nevertheless, the ability of nc886 to function as a small RNA precursor by a non-canonical pathway independent of DROSHA was recognized early [17,18,19]. Indeed, despite the removal of microRNAs derived from hsa-mir-886/vtRNA2-1/nc886 from miRBase (release v16 (11/2011)) several groups detected hsa-miR-886-3p and hsa-miR-886-5p in cells and tissues and assigned them a microRNA function in cancer. Hsa-miR-886-3p was proposed as a tumor suppressor microRNA in several types of cancer (prostate [20], bladder [21], breast [22], colon [23], lung [24,25,26,27] and thyroid [28,29]) and as a microRNA involved in other diseases (hematological [30] and Friedreich ataxia [31]). In the prostate, the loss of expression of hsa-miR-886-3p was associated with early biochemical relapse (<1 year after radical prostatectomy) [20], but a recent study shows an increased abundance of circulating hsa-miR-886-3p in high-grade compared to low-grade prostate cancer in plasma samples [32]. Notwithstanding, few reports showed an oncogenic action of hsa-miR-886-3p in renal cell carcinoma [33], colorectal carcinoma [34] and esophageal squamous cell carcinoma [35]. A recent finding proposing a nc886/PKR loss mediated doxorubicin cytotoxicity raised a novel view about its specific contribution to chemotherapy response [36]. Overall, despite of the various reports assigning a microRNA function to hsa-miR-886-3p, its production as a specific nc886 processed product, its dependence on DICER and its association to Argonaute proteins has not been consistently demonstrated. 

The aim of this study was to investigate the production of small RNAs with microRNA-like function from nc886, and the biological effect and clinical significance of this small RNAs in prostate cancer. The analysis of small-RNA-seq data from prostate cell lines and tissues demonstrated the presence of small RNAs derived from nc886 (herein designated as snc886-3p and 5p). Snc886-3p shows DICER processing hallmarks and associates with Argonaute proteins. Snc886-3p level decreases in tumor compared to normal tissues, correlating with the methylation of its gene promoter and the expression of its precursor nc886 in prostate tissue. Furthermore, we found that snc886-3p functions as a tumor suppressor microRNA-like RNA in prostate tissue, affecting cell viability, cell cycle phases and early apoptosis in PrCa cell lines. Through global gene expression analysis of cancer cell lines overexpressing snc886-3p, we identified a group of transcripts that might be targets of direct repression by snc886-3p and could explain the observed phenotype. We finally built a snc886-3p direct candidate target gene set supported by in silico predictions of microRNA binding sites AGO-PAR-CLIP (photoactivatable ribonucleoside-enhanced crosslinking and immunoprecipitation) and gene expression analysis in PRAD-TCGA and cell line data. This approach led us to the identification of 106 direct snc886-3p candidate target genes the expression of which is associated with the methylation of the nc886 promoter and a poor prognosis of PrCa patients. Overall, these findings suggest that snc886-3p functions as a non-canonical tumor suppressor microRNA derived from nc886 in prostate tissue that executes an antiproliferative effect per se, concordant with the effect of its precursor nc886.

## 2. Results

### 2.1. vtRNA2-1/nc886 Produces Small RNAs with microRNA-Like Features

To investigate the presence of small RNAs derived from nc886 in prostate tissue, we performed a re-analysis of publicly available small-RNA-seq data sets of prostate tissues and cell lines. We pooled the reads from normal and tumor cell lines transcriptomes (GSE26970, GSE66035 and SRP109305 [37,38,39]), available at Sequence Read Archive (SRA, [40]) and Gene Expression Omnibus (GEO, [41]). This analysis reveals that most of the reads mapping to nc886 gene piles up over the two sequences corresponding to the previously annotated microRNAs hsa-miR-886-3p (23 nt) and hsa-miR-886-5p (24-25 nt), both in normal (Figure 1A) and tumor (Figure 1B) cell lines. Considering the controversy of the function of hsa-mir-886/vtRNA2-1/nc886 we named the microRNA-like fragments derived from it as snc886-3p and snc886-5p. Genome BLAST analysis of the mature sequences of snc886-3p and snc886-5p shows that their only possible origin is the nc886 transcript (Appendix A). Remarkably, while snc886-3p is more abundant than snc886-5p in the normal cell lines (Figure 1A), the opposite is observed in the cancer cell lines (Figure 1B), suggesting a strand preference switch of nc886 processing upon malignant transformation. Figure 1A,B also show that the precise distribution of reads over nc886 sequence is consistent with a specific nuclease processing of nc886 instead of its random degradation [42]. Moreover, the pattern of mapping recalls the distinctive processing of DICER, defined by a sharp 5′ end cut and a less precise 3′ end cut [42,43], which is seen at both snc886-3p and snc886-5p in prostate. Consistently, a similar stacking of reads over nc886 and the same strand preference is observed in cell lines derived from different human tissues (Appendix A).

A predictive secondary structure of nc886 (101 nt) obtained using RNAstructure (MFE) essentially agrees with previous reports [45,46] (see Figure 1C, indicating the location of the two small RNAs identified in the small-RNA-seq data). The RNA folds into a hairpin (stem loop) with a 1-3 nt nucleotides overhangs in one end and an internal loop contiguous to the mature derived small RNAs, thus complying to the requirements of a DICER processing substrate [19,47]. 

A fundamental feature of a small non-coding RNA functioning as a microRNA is its association with Argonaute proteins [48,49]. To investigate if the snc886-3p/5p fragments satisfy this criterion, we re-analyzed Argonaute PAR-CLIP (AGO-PAR-CLIP) small-RNA-seq data and total small-RNA-seq of DU145 PrCa cell line (SRP075075 and SRP109305) [38,50]. We observed an enrichment of snc886-3p in the fraction associated with Argonaute proteins, similar to that observed for canonical microRNAs such as hsa-miR-301b, hsa-miR-130b and hsa-miR-320b (Figure 1D). Even though snc886-5p is more expressed than snc886-3p, it is not preferentially associated with the Argonaute fraction in DU145 cell line (Figure 1D). These findings are confirmed by a correlation analysis of AGO-PAR-CLIP versus total RNA reads of the same cell line, which shows a lower abundance but higher association with AGO-associate fraction of snc886-3p relative to snc886-5p, in the context of all the small RNAs detected in the study (Appendix A). Indeed, the association to Argonaute of snc886-3p in DU145 is above the average of microRNAs (log2 RPM Average microRNAs AGO association = 5.56, log2 RPM snc886-3p AGO association = 6.44), suggesting an active association of snc886-3p to the RISC (RNA-induced silencing complex) machinery. It is worth mentioning that experiments in other PrCa cell lines analyzed by Hamilton et al. [50] do not detect snc886-3p, probably because of its low expression (see below and Figure 2B,C) due to the high DNA methylation of its promoter [16].

Seeking additional evidence of the role of snc886-3p, we investigated its dependence on the canonical microRNA biogenesis pathway through the analysis of the changes in the small-RNA-seq transcriptome in knockouts (KO) of DROSHA, XPO5 and DICER generated in the HCT116 cell line [51]. We used a canonical microRNA (hsa-miR-130b), a Drosha-independent microRNA (hsa-miR-320b) [51] and a small RNA fragment derived from the three prime region of RNY3 RNA (sRNY3-3p) as controls. We reasoned that the global reduction in microRNA biogenesis would lead to the overrepresentation of DROSHA-independent small RNAs in the transcriptome. Thus, the observed reduction in hsa-miR-130b and increase of snc886-3p/5p and hsa-miR-320b in the DROSHA-KO indicates that the processing of the last three small RNAs is DROSHA-independent (Figure 1E). In addition, XPO5-KO produces a significant reduction in hsa-miR-130b and a small decrease in snc886-3p and snc886-5p, indicating that the two latter may be substrates of this exportin. In concordance with previous reports [12,19], DICER-KO decreases the production of snc886-3p and snc886-5p by 96% and 93% respectively, as well as the production of the control microRNAs (Figure 1E). As expected, no reduction of the fragment derived from RNY3 (sRNY3-3p) was observed for all these knockouts [52]. 

Overall, our findings indicate that nc886 is a precursor of two small RNAs that meet key microRNA criteria in prostate tissue.

### 2.2. Snc886-3p Has the Expression Profile of a Tumor Suppressor in Prostate Cells

In order to assess the relevance of snc886-3p in prostate we evaluated available microRNA expression datasets of prostate tissues (microarray and qRT-PCR). The analysis reveals a decrease in the expression of snc886-3p in cancer vs normal prostate tissue but no association with stage/prognosis (Table 1).

In addition, we analyzed the expression of snc886-3p in six paired normal and tumor samples obtained from paraffin fixed tissues derived from radical prostatectomies [16]. Quantitative RT-PCR confirmed a significant decrease of the expression of snc886-3p in tumor compared to normal tissues in this cohort (Figure 2A). Similarly, the qRT-PCR quantitation of various prostate cell lines shows a reduction in the expression of snc886-3p in tumor cell lines in comparison to the normal prostate cell line PrEc (Figure 2B). Further analysis of publicly available small-RNA-seq data of prostate cell lines (GSE26970, GSE66035 and SRP109305) reinforce this finding (Figure 2C). The reduced expression of snc886-3p in malignant vs benign cells, observed in both patient tumors and prostate cancer cell lines, is in agreement with a tumor suppressor function. 

We have previously demonstrated an epigenetically driven reduction of nc886 in PrCa, thus we wonder whether the decrease in snc886-3p is due to it. As expected, there is a positive correlation between the expression of snc886-3p and nc886 in the cell lines (r Spearman = 0.95, Figure 2D). Likewise, the expression of snc886-3p correlates with the methylation status of the 200nt region upstream to the transcription start site (TSS200nt) of nc886 (r Spearman = 0.71, Figure 2D). 

Altogether these findings suggest that the reduction of snc886-3p levels in prostate cancer is due to the decreased levels of its precursor nc886, which in turn is a consequence of the increased methylation of the nc886 promoter.

### 2.3. Snc886-3p Modulates Transcripts Affecting Cell Cycle and Apoptosis

Aiming to discover the snc886-3p effect on gene expression, we overexpressed the small RNA in DU145 cell line (Figure 3A) and performed a global gene expression analysis using Affymetrix gene microarrays 48 h after transfection. We used Sylamer algorithm [53] to estimate the enrichment of all 6-nt k-mers in the 3′ untranslated regions (3′-UTR) of transcripts of the differentially expressed genes. A landscape plot that tracks the occurrence biases of the different k-mers on all transcripts ranked by their differential expression is obtained. We observed that 5′ACCCGC3′, which is the complementary sequence to the snc886-3p seed (6-mer, 2-7-nt), is the most enriched 6-nt k-mer in the 3′-UTR of the repressed transcripts (adjusted *p*-value < 0.01) (Figure 3B). This finding favors a direct and sequence specific repressive interaction of snc886-3p with the mRNA targets. Furthermore, we identified 1358 DEGs (fold change ≥ 1.5) including 704 downregulated and 654 upregulated transcripts (Appendix A). Microarray experiments were confirmed by qRT-PCR of selected transcripts (r = 0.97, *p*-value < 0.0003) (Appendix A). Independent snc886-3p overexpression experiments in DU145, LNCaP and PC3 (Appendix A) demonstrate a similar regulation of these genes in these cell lines (Appendix A).

Since canonical microRNAs are known to act mostly through complementary nucleotide pairing at the 3′-UTR of the target gene mRNAs, we analyzed if the repressive action of snc886-3p transfection was associated with the location of its binding site along the gene. We calculated the average fold change in expression of the transcripts bearing 6-8-mers sequence complementary to the snc886-3p seed in DU145 cell line transfected with snc886-3p (microarray experiments) (Appendix A). As expected, snc886-3p causes an average global repression of the genes bearing its complementary site. Most important, its action is significantly stronger when the site is located at the 3′-UTR of the transcript (Appendix A). Furthermore, we confirmed this finding using most stringent direct target genes narrowed down for their association to AGO [50]. Transcript with a snc886-3p 6-mer (2-7-nt) binding motif at their 3′-UTR show a higher proportion of repression (39%) than those having motifs differing in two-nucleotides (24% and 22%) (Fisher’s exact test, *p*-value < 0.0001 and Odds ratio 2.0 to 5′AACCCC3′ and 2.2 to 5′AGCCAC3′ for the substituted sequences) (Appendix A). Overall, the evidence indicates that snc886-3p negative regulation of mRNA levels is dependent on the seed sequence and on the location of the complementary site in the 3′-UTR of the target genes.

In order to investigate the functional consequences of snc886-3p perturbation, we performed gene enrichment analysis of pathways and processes among the DEGs using GSEA curated gene sets [54]. The top 10 significant terms are shown in Table 2. In agreement with our previous study [16], cell cycle, apoptosis and mitogenic pathways are enriched. Particularly, three formerly validated direct targets of hsa-miR-886-3p, PLK1, TGFB1 and CDC6 [25,28], are downregulated upon snc886-3p overexpression. Additionally, candidate snc886-3p direct target CDT1, which is a partner of CDC6, has been associated with cell cycle impairment [55]. A more detailed picture of the snc886-3p responsive genes belonging to the KEGG cell cycle pathway highlights additional downregulated genes (Appendix A).

### 2.4. Snc886-3p Causes a Decrease in Cell Viability

We had previously shown that the forced recovery of nc886, the precursor of snc886-3p, reduced in vitro and in vivo cell viability of PrCa cells [16]. In view of the enrichment of cell cycle related pathways in the DEGs caused by snc886-3p overexpression (Table 2), we sought to determine if snc886-3p affects cell viability. We found that the overexpression of snc886-3p in the prostate cell lines DU145, LNCaP and PC3 produces a significant decrease in cell viability relative to the control (Figure 4A). In addition, we performed cell cycle analysis with propidium iodide to determine cells distribution in the different cell cycle phases (Figure 4B). We found an accumulation of cells in the S and G2/M phase of the cell cycle in DU145 cell line, that could be associated with the decreased expression of PLK1 evidenced previously (Appendix A). Indeed, PLK1 was reported as a target of hsa-miR-886-3p, the downregulation of which leads to an arrest in the anaphase stage of mitosis [25,56]. Additionally, LNCaP and PC3 showed an accumulation of cells in G1 phase of the cell cycle (Figure 4B). These results lead us to propose an effect of snc886-3p on apoptosis. Annexin V, a marker of early apoptosis, significantly increases in DU145 (14% of early apoptosis equivalent to an increase of 69% relative to control) and LNCaP (24% of early apoptosis equivalent to an increase of 366% relative to control) when snc886-3p is overexpressed (Figure 4C). On the contrary, we found no changes of early apoptosis in PC3 cells between conditions, though we cannot completely discard earlier apoptotic events completed at 72 h (Figure 4C). Despite the significant overall decrease in cell viability caused by snc886-3p overexpression in the three prostate cell lines tested, the distinct contribution of proliferation and apoptosis may be due to the intrinsic differences of these cell lines. Globally, these analyses show that snc886-3p decreases cell viability, due to a modulation of cell proliferation and early apoptosis.

### 2.5. Direct Candidate Target Genes of snc886-3p Are Associated with Clinical Worse Prognosis in Prostate Cancer Patients

We had previously found that nc886 [16] and snc886-3p expression (Table 1) are associated to worse clinical outcome in PrCa. We now sought to investigate if snc886-3p direct candidate target genes are also associated with the disease. A list of 253 snc886-3p possible direct target genes was built based on two criteria: the association to Argonaute determined by AGO-PAR-CLIP experiments using DU145 (reads with a sequence complementary to the 6-mer (2-7-nt) snc886-3p seed sequence) identified in Hamilton et al. [50] and the downregulation of at least 1.25 FC in DU145 after snc886-3p overexpression (Appendix A). A hierarchical clustering of the PRAD-TCGA samples in association with clinical status (pathological, clinical, Gleason score, residual tumor, biochemical recurrence) is shown in Figure 5. The expression of the 253 genes segregates the samples of patients with worse (red/violet) from those with better (non-red) clinical presentation as well as the normal tissues (green). Additionally, two major gene groups are clustered. Remarkably, the positive association of the 106 genes cluster with worse clinical parameters suggests that they may be the relevant direct candidate targets of snc886-3p repression in vivo (individual genes are listed in Appendix A). We also found that men harboring tumors with low expression of the 253 snc886-3p direct candidate target genes (percentile 25th) have a longer disease-free survival compared with the high expressing ones (percentile 75th) (HR=2, *p*-value 0.031) (Figure 6A). Disease-Free Survival analysis based on the expression of the 106-gene list showed a greater statistical difference in patient survival probability (HR=3.2, *p*-value 0.00018) compared with the 253-gene list (Figure 6A,B). Furthermore, the expression of the 106-genes associates with higher methylation of nc886 TSS200nt (thus of snc886-3p) (r 0.29, *p* < 0.0001) (Appendix A). These results provide in vitro and in vivo support of a tumor suppressor microRNA-like role of snc886-3p in prostate, acting through the direct repression of transcripts bearing complementary sequences at their 3′-UTR.

## 3. Discussion

Hsa-miR-886-3p (snc886-3p) was initially classified as a microRNA derived from pre-miR-886 [8], which was later removed from miRBase microRNA precursors (Release v16) due to its homology with vtRNA1-1/2/3 and renamed as vtRNA2-1 [9]. Although several studies reported the dysregulation of hsa-miR-886-3p in different pathological conditions [20,21,22,24,25,26,27,28,33,58], the specific synthesis of fragments with microRNA-like function from nc886 has been poorly addressed. 

We previously found that nc886 acts as a tumor suppressor in prostate cancer and the vector-driven overexpression of this RNA inhibited cell proliferation and invasion [16]. Due to existing reports about the production of small RNA with microRNA function derived from vtRNAs [59,60,61], we wonder if there is a specific processing of nc886 into small RNAs in prostate cells and if it could possibly contribute to the described nc886 induced phenotype in prostate cancer cells [16].

To assess the synthesis of vtRNA2-1/nc886 small fragments, we interrogated published small-RNA-seq prostate datasets, revealing a pattern of small RNA read stacking over nc886 sequence that resemble the cleavage of DICER which generates complementary 5p and 3p fragments (Figure 1A, Appendix A). Additionally, the analysis of Kim et al. dataset showed that DICER knockout impaired the production of snc886-3p/5p in the HCT116 cell line [51]. Interestingly, although the secondary structure of nc886 resembles a microRNA hairpin precursor, it is not fully optimized as a canonical microRNA precursor [47,62,63,64]. Indeed, the 101 nt nc886 sequence of is longer than the median 83 nt of human microRNA precursors (range 41-180 nt), generating mature small RNA fragments that are also longer (23 nt of snc886-3p and 24-25 nt of snc886-5p) than the 22 nt median human mature microRNAs (range 16-27 nt); this pattern was described for other DICER substrates of similar length [63,64]. Moreover, the presence of asymmetrical mismatches and bulges in the predictive nc886 secondary structure is associated with longer mature sequences [63]. Overall, its structural divergence with the optimized human microRNA hairpin precursor might lead to a low efficient DICER processing and consequently a small amount of microRNAs observed in our and former studies [12,18]. Evolutionary hypothesis of new microRNA precursor proposes their birth as non-efficiently processed precursor RNAs, until specific target genes are evolutionary selected; this avoids the non-specific repression of genes allowing the optimization of the hairpin structure for DICER processing [65]. The identification of the presence of a large fraction of nc886 and a less abundant fraction of derived small RNA may be the result of its recent evolutionary origin in eutherians [9]. 

To further investigate the putative microRNA-like function of snc886s we analyzed their association with Argonaute proteins. Previously, other groups reported small RNA fragments derived from vtRNA1 associated to Argonaute, holding microRNA function in breast and lymphoid tissue [59,60]. Here, we show that snc886-3p is also associated with Argonaute in lung cell line WI-38 (Appendix A), which is in agreement with the proposed microRNA identity of hsa-miR-886-3p (here snc886-3p) acting through direct repression of PLK1 and TGFB1 in lung cancer [25,27]. Concordantly, we found proof for the association of snc886-3p with Argonaute protein in prostate by analysis of AGO-PAR-CLIP available small-RNA-seq data from prostate cell line DU145 [50]. Indeed, snc886-3p enrichment is above the average of the microRNAs in these experiments, indicating a specialized microRNA-like function for this small RNA product in comparison to the 5p. Although our study shows a low association of snc886-5p to AGO it does not rule out the function of snc886-5p as a microRNA, which has been shown previously in other tissues (brain [19], cervical [66,67], breast [68]). Alternatively, snc886-5p might be involved in another pathway or be a non-functional secondary product of nc886 processing. 

In view of the possible microRNA-like identity of snc886-3p, we then investigated its association with PrCa in existing small RNA studies obtained using various techniques (qRT-PCR, microarray and small-RNA-seq). We found that hsa-miR-886-3p/snc886-3p is globally downregulated in worsened disease conditions in the published datasets (Table 1), as well as in our patient cohort (Figure 2A) and laboratory stablished cell lines (Figure 2B,C). These datasets show that snc886-3p has the expression pattern of a tumor suppressor gene the level of which in prostate cells is in the low-medium microRNAs expression range. In addition, we found that snc886-3p expression positively correlates with nc886 expression and negatively correlates with the methylation of nc886 TSS200nt. Furthermore, snc886-3p has a bimodal expression in prostate cell lines (Figure 2B,D), similar to what we reported for nc886 promoter methylation in prostate cell lines and tissues. This distribution could be explained by zygotic differences in the promoter methylation status of the gene [16]. Despite this pre-existing variation in the methylation status of nc886 TSS200nt, both patient groups present an increased promoter methylation in tumor compared with the normal matched tissue [16]. 

Different direct targets of microRNA regulation by hsa-miR-886-3p have been proposed in the literature mostly after its removal from miRBase (PLK1, TGFB1, CDC6, CXCL12 and FXN) [25,28,30,31]. If snc886-3p functions as a microRNA in PrCa it is expected to sequence-specifically bind to target mRNAs to repress their expression. In order to test this hypothesis, we performed a global gene expression study of DU145 cell line overexpressing snc886-3p vs a control small RNA. Sylamer analysis of the differentially expressed transcripts demonstrated that the sequence complementary to snc886-3p seed is the most overrepresented 6-nt k-mer in the 3′-UTR of the downregulated transcripts by snc886-3p overexpression, which strongly favors a microRNA mechanism of action. Furthermore, the identification of previously validated targets PLK1, CDC6 and TGFB1, among the downregulated transcripts of this experiment, reinforces the assumption. 

To obtain an insight of the biological effect of snc886-3p, we studied the pathways enriched in the DEGs identified by the microarray experiment, finding an enrichment in cell cycle and apoptosis processes which predicts a proliferative inhibition. This inference was confirmed by in vitro cell viability and cell cycle distribution determination in DU145, LNCaP and PC3 PrCa cell lines. Snc886-3p overexpression reduces proliferation of the three cell lines and provokes a distinctive phase specific cell cycle arrest in DU145 (G2/M) compared to LNCaP and PC3 (G1). An increase in early apoptosis after snc886-3p transfection was validated through Annexin V determination in DU145 and LNCaP, while no change was observed in PC3 cell line. Cell line specific effects may be due to differences in cell cycle control. A similar effect of snc886-3p in cell viability was previously reported in thyroid, lung and breast tissues in vitro and in vivo [22,25,27,28]. Additionally, a high-throughput analysis of seed 6-nt k-mers RNAs exposed that the snc886-3p 6-nt seed k-mer causes a significant decrease in cell viability in several non-prostate cancer cell lines [69]. In agreement with these findings, the overexpression of nc886 produces a similar phenotype in tissues where is described as a tumor suppressor (prostate [16,70], thyroid [28], gastric [71] and esophageal [72]), whereas its silencing produces the same effect in non-malignant cells in agreement with the tumor surveillance model proposed by Kunkeaw et al. (cholangiocarcinoma [73]). Accordingly, hsa-miR-886-3p and nc886 act as pro-proliferative and anti-apoptotic RNAs in tissues where it is proposed as an oncogenic RNA (renal [33], ovarian [74], thyroid [75] and endometrial [76]). Our current results together with our previous study indicate that both snc886-3p and nc886 have an antiproliferative effect and a tumor suppressor function in prostate cells. Nonetheless, given their different size, structure, and abundance, they are expected to participate in different effectors pathways. The specific production of small RNAs from nc886 is controversial in the literature, probably because of the differences in the tissue types and methods employed for its detection (RNA-seq, northern blot, microarray, qRT-PCR). In addition, the effect of both molecules has been studied using exclusively the nc886 or the snc886-3p, thus their interdependence has been mostly ignored. Indeed, the overexpression and inhibition of nc886 is expected to produce an increase/decrease of hsa-miR-886-3p/snc886-3p respectively. Meanwhile, the inhibition of snc886-3p using anti-miRs could modify the abundance, folding or endogenous interactors of the precursor molecule. As an example, a study in bladder cancer, where hsa-miR-886-3p expression is associated with short patient survival, found that both hsa-miR-886-3p knock down and nc886 overexpression lead to a decrease of cell viability in half of the cell lines tested [21]. For that reason, the effects assigned to only one of the molecules could be in fact the result of the modulation of both nc886 and its snc886 products. Indeed, the study of snc886 mimics circumvents most of these problems, allowing the discrimination of the sole effect of hsa-miR-886-3p/snc886-3p. Nevertheless, it is worth recalling that in vivo both molecules are synthesized and functional in the cells. To our knowledge, five previous reports analyzed functional effects and direct target genes of hsa-miR-886-3p/snc886-3p using mimic molecules (lung cancer) [25,27], human marrow stromal cells [30], breast cancer [22] and gastric cancer [71] and their findings are consistent to ours in prostate cancer cells. Remarkably, the only study that examined the function of both nc886 (transfection) and snc886-3p (mimic) using gastric cancer cells, found that only the transfection of nc886 (but not snc886-3p) affects cell proliferation and viability, although they showed its ability to function as a microRNA in a reporter gene assay harboring a complementary binding site at the 3′-UTR [71]. 

We and others have shown that the expression of nc886 and snc886s is associated with patient clinical outcome [20,25,71,72,74,75]. The group of Lee also showed the clinical association of the gene signature of nc886 knockout [71,72,74,75]. To our knowledge, there is no clinical association study of the snc886-3p direct candidate target genes in the literature. We built 253- and 106-gene sets composed of transcripts bearing a sequences complementary to snc886-3p seed (6-mer, 2-7-nt) protected in AGO-PAR-CLIP experiments and downregulated in our microarray experiments, the expression of which turned out to be significantly associated with a lower disease-free survival in PRAD-TCGA and unfavorable clinical parameters. Furthermore, the negative association of the 106-gene set with nc886 promoter methylation strengthens the specificity of these in vitro defined gene set. Overall, these results are strong indicators of the validity of the snc886-3p DEGs identified in vitro in DU145 cell line and reinforce the tumor suppressor function of snc886-3p in PrCa in the clinical set.

Although RNA biomarkers are increasingly established in prostate cancer management [77], snc886-3p does not fulfill relevant biomarker attributes, such us tissue specificity and upregulation in the disease condition. However, small non-coding RNAs derived from nc886 have been repeatedly detected in body fluids [78,79,80,81,82,83,84,85] and, contrary to our findings in the tumor tissue, a recent study of circulating microRNAs in the plasma of prostate cancer patients reported an increase expression of snc886-3p in high-grade compared to low-grade tumor biopsy [32]. In this context, the value of snc886-3p as a cancer biomarker is still unclear. Notwithstanding, previous reports suggested that the methylation of the nc886 promoter may be worthy of further investigation in the prostate cancer biomarker field [16,86]. Finally, the expression of snc886-3p candidate direct target genes that associate with prostate cancer prognosis may be valuable for future biomarker research.

## 4. Materials and Methods

### 4.1. Human Specimens

Tissue sections were obtained from paraffin fixed blocks stained with hematoxylin and eosin (H&E) of 6 archived radical prostatectomies and were evaluated by three pathologists at the Department of Anatomic-pathology of the Police Hospital. This study was approved by the Hospital Policial, D.N.AA.SS., Montevideo, Uruguay (2010). 

Matched normal and tumor regions, showing similar parenchyma-stroma ratio and similar cytological findings at the stroma were selected. Unstained section of 10-micron thickness, contiguous to the sections selected by the pathologist, were then freshly obtained to extract small RNAs using the RNeasy FFPE (Qiagen) Kit, with the following modifications: two extra washes with xylene and absolute ethanol were added. The RNA was resuspended in RNAse free water and stored at −20 °C for further analysis.

### 4.2. Cell Lines

LNCaP and DU145 cell lines derive from a supraclavicular lymph node and a brain metastasis respectively, while PC3 and VCaP derive from vertebral bone metastatic sites. In addition, LNCaP and VCaP are androgen sensitive, whereas DU145 and PC3 are androgen independent. Cell lines PCSC-1, PCSC-2 and PCSC-3 are highly metastatic androgen independent primary prostate cancer cell lines enriched in CD133+ stem cells. RWPE-1/2 are non-malignant cells derived from normal adult prostate epithelium and WPE-int is a less differentiated derivative of RWPE1.

LNCaP, PC-3 and DU145 human prostate cancer cell lines were obtained from ATCC (Manassas, VA, USA). LNCaP, DU145 and PC-3 were maintained in RPMI 1640 (R7755) supplemented with 10% FBS (PAA™) and penicillin/streptomycin. Cell lines PCSC-1, PCSC-2 and PCSC-3 prostate cancer stem cells (PCSCs) were purchased from Celprogen^®^; RWPE-1/2, VCaP and WPE were obtained from ATCC (Manassas, VA, USA) and maintained following the recommendations of the vendor’s specifications. All cell lines were maintained in a 5% carbon dioxide atmosphere at 37 °C. 

### 4.3. Cell Transfection

The snc886-3p mimic (5′CGCGGGUGCUUACUGACCCUU3′) and negative controls (5′UCACAACCUCCUAGAAAGAGUAGA3′) (NC, CN-001000-01-05) were synthesized Dharmacon (Miridian). According to the manufacturers’ instructions, the mimic and control mimic 20nM, were transfected into cells in 12 and 96 well plates using Lipofectamine^TM^ 3000 (Invitrogen, Carlsbad, CA, USA). After 24 h of transfection the medium was removed and fresh medium was added and 48 h (Affymetrix microarray and qRT-PCR analyses) or 72 h (Cytotoxicity assay, Flow Cytometry for DNA content analysis, Annexin V Alexa Fluor 488/PI apoptosis detection assay) after transfection cells were collected for further analysis. 

### 4.4. RNA Extraction Reverse Transcription and Quantitative Real Time PCR

Total RNA was extracted using the Qiagen^TM^ miRNAeasy kit. Quantification by reverse transcription and quantitative real time PCR (qRT-PCR) was performed using the Qiagen PCR miScript II System with oligonucleotides specific for snc886-3p (hsa-miR-886-3p (MS00010675, Qiagen)), RNAU6 (MS00033740, Qiagen) as the internal control of RNA load and the following specific gene primers pairs: 

nc886: 5′CGGGTCGGAGTTAGCTCAAGCGG3′ forward primer and 5′AAGGGTCAGTAAGCACCCGCG3′ reverse primer, as in Lee K et al. [12], PLK1: 5′CCTTGTTAGTGGGCAAACCACC3′ forward primer and 5′CGGGGTTGATGTGCTTGGGAA3′ reverse primer; TGFB1: 5′CCGTGGAGGGGAAATTGAGGG3′ forward primer and 5′GCCGGTAGTGAACCCGTTGAT3′ reverse primer; CDT1: 5′CGGAGCGTCTTTGTGTCCGAA3′ forward primer and 5′GGTGCTTCTCCATTTCCCCAGG3′ reverse primer; CDC6: 5′ATCAGGTTCTGGACAATGCTGC3′ forward primer and 5′CAATAGCTCTCCTGCAAACATCCA3′ reverse primer; C8orf82: 5′CGCGAGTATTTCTACTACGTGGACC3′ forward primer and 5′CTGCGGGTCTTTGAAGCAGGT3′ reverse primer; NLPR13: 5′CATTGCACACACTTGGGTTGGC3′ forward primer and 5′CCAGGCTCTTACTGCTGCTGAG3′ reverse primer; MMD: 5′CACACGCATTCCTCATTGTTCCG3′ forward primer and 5′TGAAGAGGGCACAGAGTCCCA3′ reverse primer; TBP: 5′GATCAAACCCAGAATTGTTCTCC3′ forward primer and 5′ATGTGGTCTTCCTGAATCCCTTT3′ reverse primer. The relative quantification was attained using the 2^−ΔΔCT^ method [87], in a Rotor-Gene 6000 equipment (Corbett Life Science). 

### 4.5. Microarray Experiments

Total RNA from 3 replicates of snc886-3p (hsa-miR-886-3p) mimic and negative controls (NC, CN-001000-01-05) was extracted after 48 h of transfection using the Qiagen^TM^ miRNAeasy kit according to the manufacturer’s protocol. The total RNA of the three replicates was pooled and labeled according to Affymetrix (Affymetrix, Santa Clara, CA, USA). Hybridization, staining and washing of the Affymetrix^®^ HG-U133 Plus 2.0 Arrays were performed with the Affymetrix Fluidics Station 450 and Hybridization Oven 640 under standard conditions (Affymetrix, Santa Clara, CA, USA). Quality control analysis and Pre-processing of the CEL-files microarray expression data was done using a graphical user interface, Chipster (v1.4.3, CSC, Finland, http://chipster.csc.fi/) following the manufacturer’s guidelines [88]. Normalization was performed using RMA algorithm [89] and annotation using the specific Affymetrix^®^ HG-U133 Plus 2.0 Arrays probe set library in Chipster. Normalized log2 expression values were used to determine the fold change expression of the snc886-3p (hsa-miR-886-3p) mimic and negative controls (NC, CN-001000-01-05) (Appendix A). 

### 4.6. Cytotoxicity Assay

At 72 h after of transfection with snc886-3p or control RNA (60-80% cell culture confluence), 20 µL of 3-(4,5-dimethylthiazol-2-yl)-2,5-diphenyl-2H-tetrazolium bromide (MTT) 5 mg/mL in 1X PBS was added to the wells and cultures were incubated for 4 h at 37 °C in a 5% CO_2_ controlled atmosphere. The medium was then aspirated and 200 µL of DMSO was added to each well and incubated at room temperature in the dark for 15 min with moderate orbital shaking. Optical density (OD) was read on a plate spectrophotometer (Varioskan^®^ Flash Multimode, Thermo Scientific, Waltham, MA, USA) at 570 nm (for formazan absorbance measurement) and 690 nm (for background measurement). 

### 4.7. Flow Cytometry for DNA Content Analysis

At 72 h after transfection with snc886-3p or control RNA (60–80% confluence) cells were harvested by trypsinization followed by two washes and resuspension in 1X PBS with gentle vortexing. Cells were then fixed by adding 1 mL of ice cold 70% ethanol dropwise and incubated at -20 °C for 30 min. Next, cells were washed with 1X PBS, centrifuged at 1200 rpm at 4 °C for 5 min and the resuspended cell pellets were incubated with 0.1 mg/mL of RNAse and 50 µg/mL propidium iodide for 15 min at room temperature in the dark. Flow cytometry measurement of nuclear DNA content was performed in an Accuri™ C6 flow cytometer (BD Bioscience), counting 10,000 total events per sample (BD Accuri C6 software). 

### 4.8. Annexin V Alexa Fluor 488/PI Apoptosis Detection Assay

Apoptosis was detected by initially staining the cells with 0.1% (*v*/*v*) Annexin V and 100 µg/mL of propidium iodide solution, according to the Annexin V Alexa Fluor 488/PI apoptosis detection assay kit (catalog n° V13241, Invitrogen, USA), followed by flow cytometry analysis in an Accuri™ C6 flow cytometer (BD Bioscience), counting 10,000 total events per sample (BD Accuri C6 software). 

### 4.9. Dataset Analysis

#### 4.9.1. Analysis of microRNA Microarray Datasets

Depending on the type of study and the availability of the data, we followed different strategies. Data deposited at GEO was analyzed using the GEO2R tool using default settings [90], selecting the samples by clinical status definition. For all the microarray data of PrCa studies analyzed, relevant features used in the analyses are listed in Table 1. 

#### 4.9.2. Analysis of Small RNA Transcriptomic Datasets

Data on microRNA expression from tumor and matched normal prostate patient samples generated by the project The Cancer Genome Atlas (TCGA) were retrieved from dbGaP, miRNAseq data Level_2 Data (file names: *.bam) of 544 samples (project approved n° 7307). Additionally, several public small-RNA sequencing expression data available at the repository Gene Expression Omnibus (GEO) [41] or Sequence Read Archive (SRA) [40] were also analyzed: microRNA expression of several human cancer cell lines (GSE16579, [91]), microRNA expression from the evaluation of the roles of DROSHA, XPO5, and DICER in microRNA biogenesis (GSE77989, [51]), AGO-immunoprecipitation of microRNAs in human senescent fibroblast WI-38 (GSE34494, [92]), microRNA transcriptome (normal prostate and prostate cells, Appendix A (GSE29904, [37])), microRNA transcriptome of DU145, LNCaP and PC3 cell lines (Appendix A (SRP109305, [38] and GSE66035, [39])), Human Prostate Cancer cell lines AGO-PAR-CLIP (SRP075075, [50]). Data was downloaded with SRA Toolkit (https://www.ncbi.nlm.nih.gov/sra/docs/toolkitsoft/) and then trimmed, mapped, annotated, counted, and normalized using miRDeep2 package software [93]. We used the mapper module (mapper.pl) with the following parameters: -e -h -l 18 -m -k “adapter-sequence” and quantifier module (quantifier.pl) with default parameters and miRBase fasta files of precursor and mature sequences from Release v21 plus manually addition of RefSeq human vtRNAs (vtRNA1-1/2/3 and 2-1) and human RNYs (RNY1/3/4/5). Normalized (Reads Per Million Reads (RPM)) log2 values were used in all cases for analysis. For the elaboration of the snc886-3p direct target candidate gene list, we used DU145 AGO-PAR-CLIP (SRP075075, [50]). Total sequencing reads were trimmed with cutadapt software [94] with the following parameters: -m 18 -q 20 and those with the sequence: 5′ ACCCGC 3′ (complementary to snc886-3p seed (6-mer, 2-7-nt)) were aligned to human genome (GRCh38/hg38) using Bowtie2 [95] with the following parameters: –L 6 -N 1. Total read counts for each gene transcript were obtained with HTSeq [96] using the Ensembl GTF file, from the reference genome sequence (GRCh38.97). 

#### 4.9.3. Analysis of Methylation Microarray Datasets

The methylation data of the PRAD-TCGA cohort, was extracted from the Illumina Infinium Human Methylation 450 BeadChip array data of the 49-paired normal and prostate tumor samples and additionally unmatched normal and tumor tissues (336 in total). Additionally, public methylomes available at the repository Gene Expression Omnibus (GEO) [41] obtained using Illumina Infinium Human Methylation 450 BeadChip arrays were also analyzed prostate cell lines PrEc, RWPE1, VCAP, LNCaP, DU145 and PC-3 gene dataset GSE68379 [97]. The average of the normalized beta-values for the 6 CpGs sites located at the nc886 TSS200nt promoter (cg18678645, cg06536614, cg26328633, cg25340688, cg26896946, cg00124993) were calculated. 

#### 4.9.4. Heatmap of Hierarchical Clusterization of 253 snc886-3p Candidate Direct Target Genes

Heatmap was performed by two-way hierarchical clustering using the Spearman rank correlation algorithm using Morpheus (https://software.broadinstitute.org/morpheus/) and gene expression values and clinical status for different parameters (Clinical T value, Pathological T value, Gleason Score, Residual Tumor and Biochemical Recurrence) of PRAD-TCGA dataset. 

### 4.10. Statistical Analysis

All experiments were performed at least in triplicate and the corresponding variables are expressed as average value ± standard deviation or standard error (referred in the figure). Statistical analyses were done using single, two-tailed t-test, one-way and two-way ANOVA for multiple comparison tests, including Tukey’s Honest Significant Difference test as a post-hoc tests (referred in the figure). D’Agostino–Pearson was conducted as normality test and Pearson or nonparametric Spearman was used to test correlation. Two-tailed Fisher exact test for difference in the proportions of genes was used. All the analyses were done in GraphPad Prism 6. The observed differences were expressed using the *p*-value (* *p* < 0.05, ** *p* < 0.01, *** *p* < 0.001, **** *p*-value < 0.0001). Results with a *p*-value of < 0.05 were considered significant.

## 5. Conclusions

Our study demonstrates the presence of hsa-miR-886-3p/snc886-3p in prostate tissue derived from the DICER mediated processing of vtRNA2-1/nc886, which associates to argonautes repressing transcripts bearing complementary seed sequences, thus functioning as a microRNA. We also found a DNA methylation dependent downregulation of snc886-3p and a concomitant upregulation of direct candidate targets of repression, both associated with PrCa disease condition and progression. Snc886-3p effects on global gene expression support the modulation of cell cycle progression and apoptosis observed in prostate cancer cell lines. Altogether, our results indicate that both nc886 and snc886-3p are simultaneously expressed in prostate cells at different levels, exerting a tumor suppressor action through probably different effector pathways.

## Figures and Tables

**Figure 1 ncrna-06-00007-f001:**
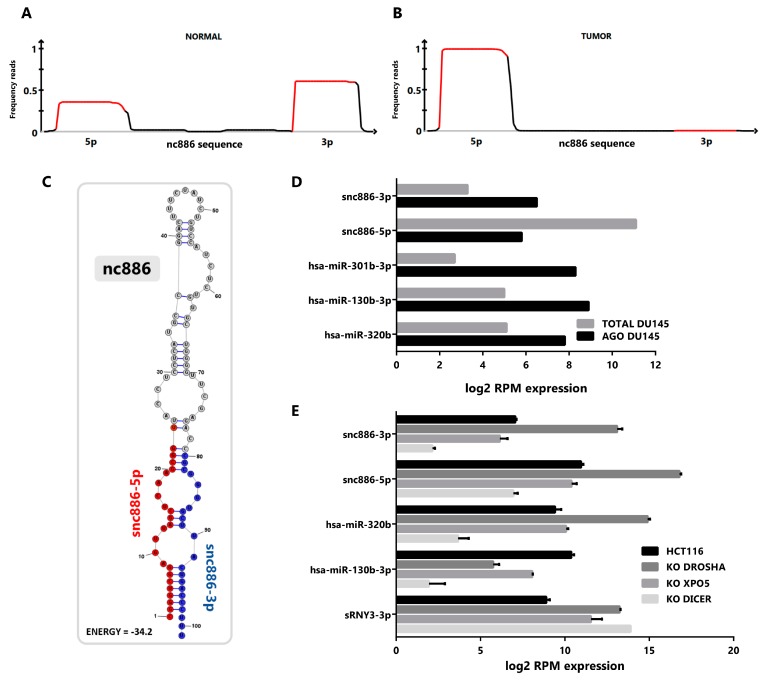
nc886 derived fragments are produced in prostate cell lines and exhibit microRNA features. (**A**,**B**) Mapping of prostate cell small RNA reads along the sequence of nc886. Reads obtained from small-RNA transcriptomes of non-transformed prostate PrEc and PrSC (GEO id: GSE26970) (**A**) and tumor DU145, PC-3 and LNCaP cell lines (SRA id: SRP109305 and GEO id: GSE66035) (**B**) were mapped to nc886. A diagram of nc886 (Refseq (NR_030583.2)), plus additional 10 nt at both ends, is depicted below the plots. Red lines in the plots represent the previously annotated hsa-miR-886-3p and -5p. (**C**) Predicted secondary structure of nc886 (Refseq (NR_030583.2)) based on maximum free energy (MFE) generated with the RNAstructure software [44]. (**D**) Association of small RNAs to Argonaute in transcriptomic analysis of DU145 cell line. The graph shows the normalized expression, reads per million (RPM), of small non-coding RNAs in total cellular RNA (TOTAL DU145) and in the Argonaute PAR-CLIP fraction (AGO DU145). Data set available at SRA id: SRP075075. (**E**) Transcriptomic analysis of small non-coding RNAs of the HCT116 cell line with knockouts for the microRNAs biogenesis proteins DROSHA, EXPORTIN 5 and DICER. Data set available at GEO id: GSE77989. Average value and standard error are shown.

**Figure 2 ncrna-06-00007-f002:**
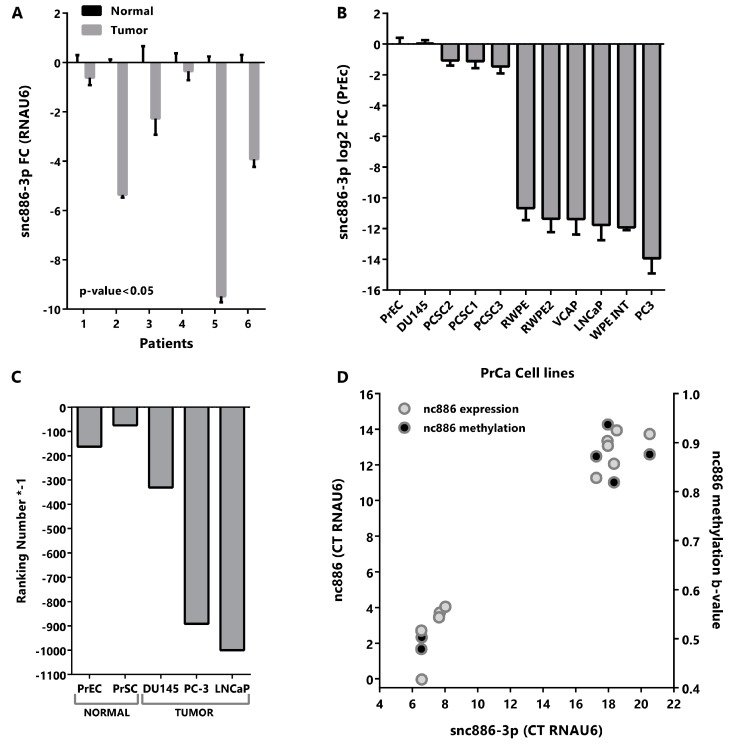
Snc886-3p is downregulated in tumor vs normal prostate tissue and cell lines. (**A**) Expression of snc886-3p in paired normal and tumor tissues from patient prostatectomies. Expression of snc886-3p in tumor relative to normal matched prostate tissue was determined by qRT-PCR (ΔΔCt) using RNAU6 as normalizer. Two-tailed paired T-test was performed. (**B**) Expression of snc886-3p in prostate cell lines. Expression of snc886-3p relative to normal prostate cell line PrEc, determined by qRT-PCR (ΔΔCt) using RNAU6 as normalizer. The order of the cell lines is only based on the expression of snc886-3p (**C**) Expression of snc886-3p in transcriptomic analysis of small non-coding RNAs from different prostate normal (GEO id: GSE26970) and tumor (SRA id: SRP109305 and GEO id: GSE66035) cell lines. For a better comparison of the different cell lines considering the differences in the depth sequencing among experiments, ranking number of appearances of snc886-3p is shown. (**D**) Correlation between nc886 and snc886-3p expression and average TSS200nt methylation on prostate cell lines (GSE68379: PrEc, RWPE-1, DU145, PC3, VCAP and LNCaP). Expression data is the same as in (**B**). Average values and standard deviation are shown.

**Figure 3 ncrna-06-00007-f003:**
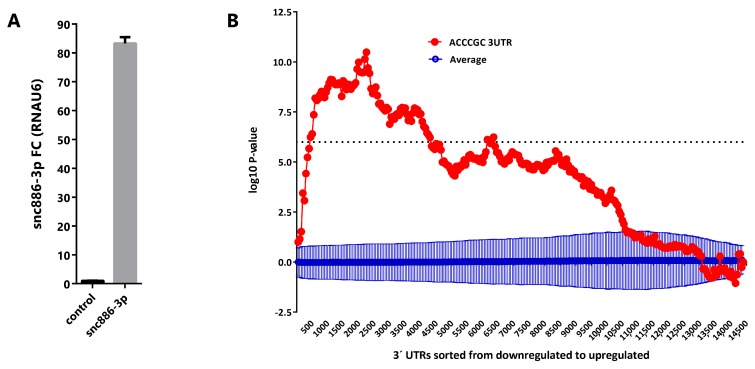
Snc886-3p downregulates transcripts bearing a 5′ACCCGC3′ sequence complementary to its putative seed in their 3′-UTR. (**A**) Expression of snc886-3p in DU145 cells transfected with 20nM of mimic snc886-3p and negative control (Dharmacon) after 48 h. Quantification by qRT-PCR using RNAU6 as a normalizer. (**B**) Sylamer enrichment landscape plot for 6-nt k-mer in all genes ranked by their change in abundance after snc886-3p overexpression compared to control determined by Affymetrix microarrays. The *x*-axis represents the sorted gene list (downregulated and upregulated genes are plotted in the left and right part of the axis respectively). The *y*-axis shows the hypergeometric significance for each 6-nt k-mer at each leading bin. Positive and negative values indicate enrichment or depletion of the 6-nt k-mer sequence at the 3′-UTR of the genes. The dotted horizontal line symbolizes an E-value threshold of 0.01 (Bonferroni corrected). The red line embodies the values obtained for the 6-nt k-mer (5′ACCCGC3′) complementary to the 6-nt seed (6-mer, 2-7-nt) of snc886-3p. The blue line shows the average profile of 6-nt k-mer unrelated to the seed region of snc886-3p (dark blue), with standard deviation as vertical bars.

**Figure 4 ncrna-06-00007-f004:**
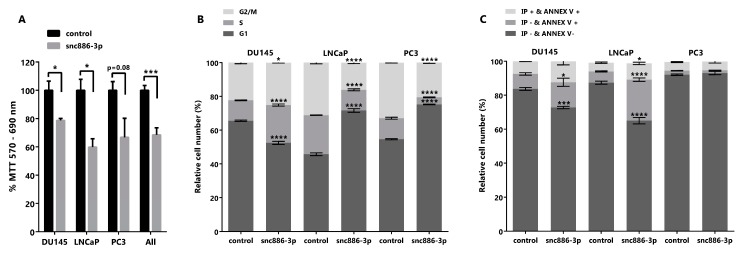
Cytotoxicity, cell cycle and early apoptosis assay in prostate tumor cell lines overexpressing snc886-3p. DU145, PC-3 and LNCaP cells transfected with 20 nM of mimic snc886-3p and negative control (Dharmacon) were evaluated 72 h after transfection. All analyses were performed in triplicates to determine average values and standard error. * *p*-value < 0.05; *** *p*-value < 0.001; **** *p*-value < 0.0001. (**A**) MTT assay to determine cell viability. *p*-values of T-test of the individual cell lines and the three cell lines considered as replicates (all) are indicated over the horizontal brackets. (**B**) DNA content analysis to determine the distribution of cell cycle stages. 10,000 cells/experiment were stained with Propidium Iodide and evaluated by flow cytometry. (**C**) Early apoptosis assay. 10,000 cells/experiment were stained with Annexin V (Annex V) and Propidium Iodide (IP) and evaluated by flow cytometry. Viable cells are negative for Annexin V and Propidium Iodide, cells positive for Annexin V and negative for Propidium Iodide are in early apoptosis and cells positive for Annexin V and Propidium Iodide are necrotic cells. Two-way ANOVA Test for the cell cycle and early apoptosis assays was used to estimate statistical significance of the observed differences.

**Figure 5 ncrna-06-00007-f005:**
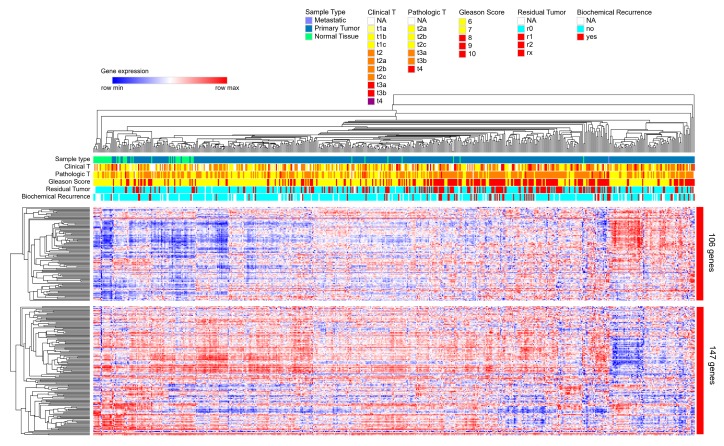
Hierarchical clustering based on the expression of 253-snc886-3p direct candidate target genes in prostate tissue of PRAD-TCGA. Heatmap of the expression of the 253 genes obtained by a two-way hierarchical clustering using the Spearman rank correlation algorithm with the Morpheus software and PRAD-TCGA data. The principal two gene clusters formed are shown (cluster of 106 genes and cluster of 147 genes). Color scale indicates the relative expression level of the genes across the samples (red and blue represent higher and lower expression relative to the mean). Clinical status for different parameters (Clinical T value, Pathological T value, Gleason Score, Residual Tumor and Biochemical Recurrence) are represented by horizonal marks on the top of the heatmap. The values for each parameter are indicated by colors in the top right legends.

**Figure 6 ncrna-06-00007-f006:**
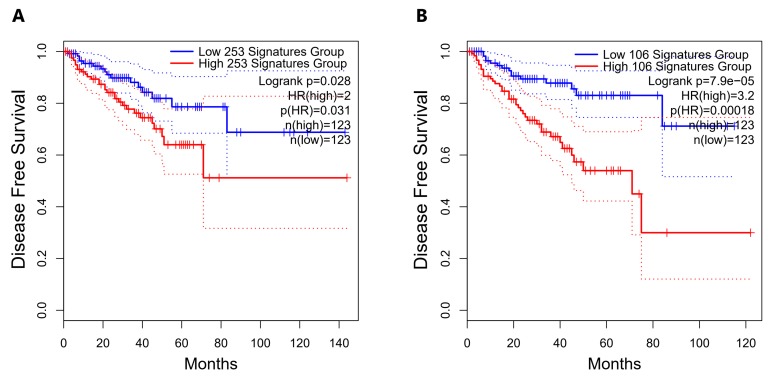
Disease-Free Survival analysis for PRAD-TCGA patients according to the expression of the snc886-3p direct candidate target genes. Disease-Free Survival Kaplan–Meier curve for the expression status of candidate target genes of snc886-3p in PRAD-TCGA was generated using GEPIA2 software [57]. (**A**) 253-snc886-3p direct candidate target gene list. (**B**) 106-snc886-3p direct candidate target gene list. Two groups of patient tumors are compared: percentile 25th expression of the candidate target genes (low expression: blue line) and percentile 75th expression of the candidate target genes (high expression: red line). Hazard Ratio (HR), *p*(HR) and number of patients in each group are indicated. The dotted lines represent the confidence interval of each group.

**Table 1 ncrna-06-00007-t001:** Analysis of snc886-3p abundance in PrCa microRNA datasets.

GEO ACC *	PUBMED ID	Platform	Total Samples	Benign	Cancer	Fold Change (N/B vs T)	*p*-Value	Analytical Tool
GSE45604	24518785	Affymetrix	60	10 Normal	50 Cancer	−1.4	0.05	GEO2R
GSE26964	21647377	Capitalbio	13	6 Primary PrCa	7 Metastasis Bone	−2.7	N.S.	GEO2R
GSE23022	21400514	Affymetrix	40	20 Normal	20 Cancer	−1.3	0.03	GEO2R
GSE55323	24967583	Agilent	40	20 Non-Recurrent	20 Recurrent	−1.2	N.S.	GEO2R
GSE62610	25416653	Taqman qPCR	36	14 Non-Recurrent	22 Recurrent	−1.7	N.S.	GEO2R
GSE21036	20579941	Agilent	140	28 Normal	112 Cancer	−2.0	0.001	GEO2R
GSE36802	23233736	Affymetrix	42	21 Benign	21 Cancer	−1.8	0.0002	GEO2R
TCGA data	26544944	Small-RNA-Seq	24 ^a^	12 Normal	12 Cancer	−3.2	0.03	miRDeep2

* GEO accession number; ^a^ PRAD-TCGA paired normal/tumor samples (52) with at least 2 reads for sdnc886-3p in normal tissue (24), paired t-test; N.S. non-significant.

**Table 2 ncrna-06-00007-t002:** Gene enrichment analysis GSEA (curated gene sets).

Gene Set Name	Genes in Gene Set (K)	Genes in Overlap (k)	*p*-Value	FDR *q*-Value
KEGG_NEUROTROPHIN_SIGNALING_PATHWAY	126	21	1.26 × 10^−9^	2.34 × 10^−7^
KEGG_INSULIN_SIGNALING_PATHWAY	137	18	7.63 × 10^−7^	6.13 × 10^−5^
KEGG_APOPTOSIS	88	14	1.26 × 10^−6^	6.13 × 10^−5^
KEGG_CELL_CYCLE	128	17	1.32 × 10^−6^	6.13 × 10^−5^
KEGG_PATHWAYS_IN_CANCER	328	28	6.45 × 10^−6^	2.40 × 10^−4^
KEGG_VALINE_LEUCINE_AND_ISOLEUCINE_DEGRADATION	44	9	1.22 × 10^−5^	3.79 × 10^−4^
KEGG_TGF_BETA_SIGNALING_PATHWAY	86	12	2.81 × 10^−5^	6.73 × 10^−4^
KEGG_CHRONIC_MYELOID_LEUKEMIA	73	11	2.89 × 10^−5^	6.73 × 10^−4^
KEGG_MAPK_SIGNALING_PATHWAY	267	23	3.61 × 10^−5^	7.46 × 10^−4^
KEGG_MTOR_SIGNALING_PATHWAY	52	9	4.99 × 10^−5^	9.28 × 10^−4^

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
