# Peer review of "vtRNA2-1/nc886 Produces a Small RNA That Contributes to Its Tumor Suppression Action through the microRNA Pathway in Prostate Cancer"

_ncrna, 2020, doi:10.3390/ncrna6010007_

Round 1

Reviewer 1 Report

This article provides useful insights into the functional role of nc866 in prostate cancer. Before publication, I would suggest the following amendments:

Please read carefully the text, there are several typos/errors (e.g. line 43 “Although the… [Despite]”; line 47 “not function has.. [no function]”) Line 108: can the BLAST analysis data be shown (perhaps as supplementary info)? The interpretation of Table 1 is problematic: most prognostic data are not significant (met vs non-met; recurrent vs non-recurrent). Hence the authors should change their description of the table: the results show a clear reduction in cancer vs normal, but no association with stage/prognosis etc. Table 1: what is the difference between “benign” and “normal”? If they are the same, please use just one term for clarity.  Figure 2A: can the authors please specify the statistical analysis performed (paired or unpaired t test?) and justify this decision? Figure 4 B and C: I see many significant results, even for differences that are quite small. This could be due to the fact that the authors seem to have used ANOVA with no post-hoc test. Can the authors please perform a more appropriate analysis, or justify their choice?

Reviewer 2 Report

The paper by Fort et al describes the production of a small RNA with microRNA-like function, nc886, which is capable of acting as a non-canonical tumor suppressor in prostate cancer. Work presented follows from recent studies in the literature, and the authors’ own work, where nc-886 has been described as an important molecular signal in cancer tissue. Processing and function of different small nc transcripts is very relevant to our understanding of ncRNA biology. The same transcript can have intrinsic biological function in addition to the further processed products. The originality of the current work lies in the confirmation of micro-RNA like properties for the 3p fragment of nc-886, a functional mechanism that has been largely overlooked since removal of nc-886 from list of potential miRNAs.

The study is well designed and presented, with clear experimental approaches and analytical processes. Correlation analysis with clinical prognosis is quite demonstrative of the utility of transcriptomics, and big data in general, and of the importance of its free public availability to the scientific community. The premise validated ads novelty to our current understanding of nc-886 function and could potentially have wider implications for other ncRNAs. As such and provided some relatively minor edits and comments are followed, I can recommend its acceptance for publication in non-coding RNAs.  

There are a few points where more detail and descriptive parameters could be added (see below for specific Results comments). The Discussion could have a greater scope, briefly including some recent description of nc-886 molecular mechanisms. For example, the paper by Kunkeaw et al (2018), on mechanism mediated by nc-886 should be mentioned and discussed in the context of the authors’ findings.

Results comments

Line 214: briefly describe DU145. In table 1 and Figure 2 are cell lines/tissues displayed according to tumorigenic capacity? Figure S2C might be better placed next to the data in Table 2.

Discussion comments

Considering the findings of this study, it is important that the authors briefly discuss past literature where functional mechanisms are ascribed to mir-866-3p. For example, Xiong et al. (2011) do use a precursor rather than a mature mimic for overexpression studies, meaning that the effect could be theoretically both miRNA- dependent and -independent. Are there any past studies with mature/mimic overexpression of miR-886-3p?

It would be interesting if the authors could discuss the potential role of snc-886-5p, given its lack of association with argonaute fraction in DU145, and its presence in PAN-AGO fractions in WI-38.

Taking into consideration the correlation with patient prognosis, can the authors speculate about the biomarker potential of nc-886-3p? and its target genes.

Line 340: please cite some of those papers

Minor edits

Line 47: typo in “not”.

Line 58-62: The authors should mention that this is their previous past work. As this is not immediately obvious from the text.

Line 62: “small RNA”

Line 352: substrates

Line 370: Please replace “of the” for “for the”.

Line 560: Please replace “evidences” with “demonstrates” or similar.

Line 442: If possible please provide the sequence for the snc886 mimic and negative controls.

Reviewer 3 Report

In the proposed manuscript the authors investigated the production of small RNAs with microRNA-like

Function processed from the vault RNA nc886, then they explored the biological effects and clinical significance of these small RNAs in prostate cancer.

The analysis of small-RNA-Seq public data, from prostate cell lines and tissues, demonstrated the presence of small RNAs derived from nc886 designated by the authors as snc886-3p and 5p. While both small RNAs showed Dicer processing hallmarks, only Snc886-3p was found associated with Argonaute proteins in DU145 cell line, thus prompting the authors to suppose for this a miRNA like activity. Through snc886-3p overexpression, they demonstrated its role in affecting cell viability, cell cycle phases and early apoptosis in prostate cancer cell lines. Finally, the authors identified 106 genes candidate as direct snc886-3p  targets, whose expression is associated with the methylation of the nc886 promoter and poor prognosis of PrCa patients. Overall, the presented data suggest that snc886-3p functions as a non-canonical tumor suppressor microRNA.

The main topic is very interesting but some aspects might be further investigated in order to give more robustness to the presented data.

The introduction is exhaustive.

The authors investigated if snc886-3p overexpression might affect cell viability and cell cycle in the prostate cell lines DU145, LNCaP, and PC3. Despite snc886-3p overexpression has induced a significant decrease in cell viability in the three prostate cell lines, cell cycle and apoptosis analyses revealed a different response to ncRNA expression. It is possible that the differences between cell lines may have been amplified by the timing chosen for the analysis.

I would suggest repeating the analyses at a shorter time, i.e. 24 and 48 hours after transfection so that the effects directly attributable to snc886-3p expression can be appreciated. The analyses performed 72 hours after transfection might be the result of a cascade of molecular events in which the intrinsic differences among cells can play an important role.

In addition, even if bioinformatic analyses supported the hypothesis of snc886-3p as non-canonical micro RNA, I suggest to formal proof in cells the miRNA like activity by the use of miRNA target vectors on a couple of the genes identified as miRNA targets,  this would give more robustness to the data. 

Round 2

Reviewer 3 Report

The manuscript has been improved by the revision.